# Proteome Landscape during Ripening of Solid Endosperm from Two Different Coconut Cultivars Reveals Contrasting Carbohydrate and Fatty Acid Metabolic Pathway Modulation

**DOI:** 10.3390/ijms241310431

**Published:** 2023-06-21

**Authors:** Jean Wildort Félix, María Inés Granados-Alegría, Rufino Gómez-Tah, Miguel Tzec-Simá, Eliel Ruíz-May, Blondy Canto-Canché, Jesús Alejandro Zamora-Briseño, Esaú Bojórquez-Velázquez, Carlos Oropeza-Salín, Ignacio Islas-Flores

**Affiliations:** 1Unidad de Bioquímica y Biología Molecular de Plantas, Centro de Investigación Científica de Yucatán, A.C., Calle 43 No. 130 x 32 y 34, Chuburná de Hidalgo, Mérida C.P. 97205, Yucatán, Mexico; felixjeandewildort@yahoo.fr (J.W.F.); granados.alegria@gmail.com (M.I.G.-A.); tzecmyr@cicy.mx (M.T.-S.); 2Unidad de Biotecnología, Centro de Investigación Científica de Yucatán, A.C., Calle 43 No. 130 x 32 y 34, Chuburná de Hidalgo, Mérida C.P. 97205, Yucatán, Mexico; rufino26gt@gmail.com (R.G.-T.); cantocanche@cicy.mx (B.C.-C.); cos@cicy.mx (C.O.-S.); 3Red de Estudios Moleculares Avanzados, Instituto de Ecología, A.C., Carretera antigua a Coatepec 351, Colonia El Haya, Xalapa C.P. 91073, Veracruz, Mexico; eliel.ruiz@inecol.mx (E.R.-M.); alejandro.zamora@inecol.mx (J.A.Z.-B.); esau.bojorquez@inecol.mx (E.B.-V.)

**Keywords:** proteomics, coconut cultivars, solid endosperm, maturation stages, carbohydrate and lipid metabolisms, TMT-SPS-MS3

## Abstract

*Cocos nucifera* L. is a crop grown in the humid tropics. It is grouped into two classes of varieties: dwarf and tall; regardless of the variety, the endosperm of the coconut accumulates carbohydrates in the early stages of maturation and fatty acids in the later stages, although the biochemical factors that determine such behavior remain unknown. We used tandem mass tagging with synchronous precursor selection (TMT-SPS-MS3) to analyze the proteomes of solid endosperms from Yucatan green dwarf (YGD) and Mexican pacific tall (MPT) coconut cultivars. The analysis was conducted at immature, intermediate, and mature development stages to better understand the regulation of carbohydrate and lipid metabolisms. Proteomic analyses showed 244 proteins in YGD and 347 in MPT; from these, 155 proteins were shared between both cultivars. Furthermore, the proteomes related to glycolysis, photosynthesis, and gluconeogenesis, and those associated with the biosynthesis and elongation of fatty acids, were up-accumulated in the solid endosperm of MPT, while in YGD, they were down-accumulated. These results support that carbohydrate and fatty acid metabolisms differ among the developmental stages of the solid endosperm and between the dwarf and tall cultivars. This is the first proteomics study comparing different stages of maturity in two contrasting coconut cultivars and may help in understanding the maturity process in other palms.

## 1. Introduction

The coconut tree (*Cocos nucifera* L.) is an ancient palm species of great importance in tropical and subtropical areas [1]. Coconut belongs to the Arecaceae family, which also includes the oil palm (*Elaeis guineensis*) and the date palm (*Phoenix dactylifera* L.); these palms are sources of oil (the former) and carbohydrates (the latter) for the food and pharmaceutical industries [2,3].

The coconut tree shows many morphological and genetic variations, e.g., in tree and fruit sizes, as well as flower pollination, but coconuts are classified into two groups based on morphology and growth habits: the “tall” and the “dwarf” varieties [1,4]. Additionally, hybrids exist, which result from cross-pollination between a tall and a dwarf variety [5,6]. Tall varieties begin to flower 5–7 years after planting and continue to produce inflorescences and fruits until they are 80 to 100 years old. As adults, the tall palms reach 20–30 m in height, producing medium- to large-sized fruits with abundant solid endosperm and high oil content [7]. In the case of dwarf palm varieties, they begin to flower and fruit 3–4 years after planting and continue to produce until they are 50 years old. As adults, they are 8–10 m tall and produce small to medium-sized fruits with moderate amounts of solid endosperm, but lower oil content than tall varieties [8]. The seed of the coconut tree is a drupe; its shape varies from spherical to oval and includes the solid endosperm, the embryo embedded in the solid endosperm, and the slightly sweet liquid endosperm (coconut water) [9].

The solid endosperm, also called “coconut meat”, and when dry, “copra”, is a source of oil composed of triacylglycerol (TAG) with long-, medium- and short-chain fatty acids. In the solid endosperm, medium-chain saturated fatty acids (MCSFA) predominate, e.g., lauric acid (C12:0), myristic acid (C14:0) and palmitic acid (C16:0), among others, and the total fatty acid content is higher in the tall than in the dwarf variety [10,11]. Coconut oil contains high levels of lauric acid and exhibits characteristics such as increased oxidative stability, low melting points and stable emulsion formation, all highly appreciated in the food and chemical industries [12,13].

The growth and maturation of coconut fruit are intrinsically related to the development of the different components of the seed, i.e., endosperms, pericarp and embryo. However, in contrast to the liquid endosperm, the solid endosperm of the coconut fruit does not accumulate synchronically with fruit maturation because it begins to accumulate around the internal periphery of the embryo sac sixth months after pollination; early solid endosperm cells have an appearance of a cream to transparent jelly-like matrix and contain many kinds of structural carbohydrates [14]. When fruits are ripening, the solid endosperm becomes thicker, white in color, with a more compact structure; then, endosperm cells stop their deposition and begin to function as a reservoir of proteins and triacylglycerols [14].

Bourgis et al. [15] analyzed the transcriptomes of oil and date palms, two Arecaceae species with contrasting metabolisms. They found that the high oil content in *E. guineensis* was associated with high transcriptional expression levels of fatty acid biosynthesis genes, while transcriptional levels of glycolysis enzymes were comparable in both species. Naganeeswaran et al. [16] performed transcriptomics analysis of *P. dactylifera* embryogenic calli, identifying 53,251 transcripts. The KEGG-enriched pathways were predominantly represented by metabolic pathways that include carbohydrate and energy, consistent with the almost exclusive accumulation of carbohydrates in date palm. In the case of coconut palm, the seed accumulates carbohydrates in the early stages of development, while in the intermediate and mature stages, it accumulates fatty acids [5]. Transcriptomics analysis is highly valuable, but transcript expression does not necessarily correlate with protein translation; thus, proteomics analysis is also necessary.

Regarding proteomics studies carried out on coconut solid endosperm, D’Amato et al. [17] reported a proteome of 307 proteins associated with coconut milk, a derivate of the solid endosperm. Huang et al. [18], using two-dimensional electrophoresis (2-DE), generated a proteomic map of common proteins in the solid endosperms of 14 coconut samples collected in different provinces of China, Thailand and Vietnam. They found at least 200 protein spots, and sequencing some of these proteins by MALDI-TOF-TOF-MS revealed homology to 7S globulin, glutelin and a putative receptor-like protein kinase. Recently, Ma et al. [19], using shotgun protein sequencing of coconut solid endosperm, identified 1686 proteins. The in silico digestion of globulins and antioxidant proteins with pepsin and trypsin helped them to predict peptides with emulsifying and antioxidant activities. Unfortunately, the database for this research is not publicly available, preventing further analysis. Moreover, all these investigations seem incomplete because they omit information about the varieties and developmental stages of the fruit used, since proteomic changes are influenced, at least in part, by these factors.

Coconut is a non-climacteric fruit with only a few studies focused on biochemical maturation processes. The Yucatan green dwarf (YGD) and the Mexican pacific tall (MPT) are two coconut cultivars with contrasting characteristics; the fruit of the former is used mainly for water consumption, while the second is mainly used for copra production and oil extraction [6]. However, the molecular and biochemical basis for these differences have not been revealed yet.

This study aims to investigate the regulation of carbohydrate and fatty acid metabolisms during coconut fruit ripening and uncover the proteins involved in these metabolic processes in coconut fruit. Using tandem mass tag–synchronous precursor selection (TMT-SPS-MS3), proteomic analysis was carried out on the solid coconut endosperm, comparing the meat of immature, intermediate and mature fruits of the Yucatan green dwarf (YGD) and the Mexican pacific tall (MPT) cultivars.

A total of 244 proteins were identified in YGD and 347 in MPT cultivars; enzymes involved in lipid and carbohydrate metabolism were found in both cultivars. Proteins involved in glycolysis and methylglyoxal pathways were up-accumulated in MPT but down-accumulated in YGD, consistent with their roles in lipid accumulation in other plants. Regarding lipid metabolism, the proteins showed higher accumulation in MPT than in YGD as maturation progressed, which corroborates that the MPT cultivar is best suited for oil extraction because of their high TAG content.

To the best of our knowledge, this is the first comprehensive proteomics report focusing on the maturity process of solid coconut endosperms of two contrasting varieties, thus contributing to a better understanding of lipids and carbohydrate metabolisms in coconut, but also in other oilseeds, which are poorly studied in terms of their maturation processes.

## 2. Results

### 2.1. Physiological and Physicochemical Parameters of the Coconut Fruit

Our study was centered on three coconut ripening stages: immature, intermediate and mature fruits of the Yucatan green dwarf (YGD) and Mexican pacific tall (MPT) cultivars (Figure 1A,B). In both cultivars, the main morphological characteristics of coconut fruits during ripening were associated with changes in color and weight. At the immature stage, the fruits are green, while at the intermediate, they are yellowish-green, and finally, they are pale brown at the mature stage. The color modifications were accompanied by the drying of the pericarp, a decrease in coconut water volume, and a loss of weight of mature fruits (Figure 1A,B and Appendix A).

Both cultivars exhibited similar morphological features during solid endosperm ripening (Figure 1A,B). The immature solid endosperm was gelatinous and slightly transparent to whitish in color. In the intermediate stage, the endosperm was thicker and harder than in the immature fruit. In the mature stage, the solid endosperm was white, thicker and harder than in the previous stages (Figure 1A,B).

Complex protein profiles were observed in a 12% SDS-PAGE for each of the different stages of maturity and in both cultivars, with observed molecular sizes between 17 to 199 kDa. Four major polypeptides with molecular weights of 46, 33, 24.8 and 17 kDa were present in all stages of both cultivars, but particular differences were also observed. A 30.7 kDa polypeptide was present in all three stages of maturation in the YGD cultivar, whereas it was only observed in the immature stage of the MPT cultivar. A 20.7 kDa polypeptide was more abundant in the mature stage in the YGD cultivar (Figure 1C). The comparative-proteomics-based TMT-SPS-MS3 allowed the identification of 244 and 347 proteins in YGD and MPT cultivars, respectively (Figure 1D,E and Appendix A). Both cultivars shared 155 proteins, indicating that these proteins were the core of the proteome, whereas 89 proteins were unique to YGD and 192 were unique to MPT (Figure 1E).

### 2.2. Proteome Landscape of Coconut Solid Endosperm during Ripening

The core clusters of the MPT and YGD cultivars were visualized in TreeMap to analyze the enriched metabolic pathways. The glycolytic process, including malate metabolism, protein folding, proteasomal ubiquitin independent protein catabolism, together with the stress response, were the main biological processes enriched in both cultivars (Figure 2A–C and Appendix A). In addition, YGD exhibited an enrichment of fatty acid biosynthesis, protein refolding, and embryo development ending in seed dormancy (Figure 2B). The MPT super cluster was particularly enriched with proteins associated with biological processes such as the tricarboxylic acid cycle, the glyoxylate cycle, citrate and malate metabolism, tetrahydrofolate interconversion, chlorophyll biosynthesis, photosynthesis, cytoskeleton organization, and protein transport (Figure 2C).

### 2.3. Dynamics of the Coconut Solid Endosperm Proteome during Ripening

The normalized protein abundance of the biological replicates originating from the immature, intermediate and mature stages was analyzed by principal component analysis (PCA). The biological replicates under study were grouped together according to each stage of maturation and the cultivar (Figure 3A,E). In the YGD cultivar, the accumulated variance of PC1 (59.8%) and PC2 (25.7%) explained up to 85.5% of the variation observed between the ripening stages analyzed (Figure 3A); while in MPT, the accumulated variance of PC1 (72.1%) and PC2 (18.4%) explained up to 90.5% of the variation observed there (Figure 3E). An additional PCA was carried out to compare the two cultivars and their stages of maturity between them; the PCA shows differences between cultivars and among their different stages of maturation (Appendix A).

Subsequently, the trend in protein accumulation was analyzed during coconut ripening, focusing on the ratios intermediate/immature (Int/Imm), mature/immature (Mat/Imm), and mature/intermediate (Mat/Int). According to the log2-fold change (log2FC), a total of 225 differentially accumulated proteins (DAPs) were found in YGD, whereas 251 DAPs were found in MPT (Appendix A). With respect to the Mat/Int in YGD, proteins like putative FAM10 family, ANXD1 and EF1B1 were down-accumulated, while ALAT2, EDA2 and GEP148 were up-accumulated. When comparing the Mat/Imm and Mat/Int of YGD coconut meat, HSP90-4, BCCP, TCPQ, DIS3, LPD2, PFK6 and ALAT2 were down-accumulated in the mature stage, while proteins like Enod1, AGAL2, GEA6 and SCII-3 were up-accumulated (Figure 3B–D). In the MPT coconut cultivar, SUS4 and SPD6 were down-accumulated in the intermediate and mature stages, while the GbG protein increased during maturation and, finally, LEA was up-accumulated in the later stages (Figure 3F–H).

### 2.4. Detoxification of ROS and Methylglyoxal Pathways in MPT and YGD Cultivars

ROS detoxification was more active in MPT than in YGD. Proteins such as superoxide dismutase (P24704, O81235) and catalase (P25819) were significantly up-accumulated in the intermediate and late stages of MPT maturation (Figure 4). Furthermore, the glutathione S-transferase DHAR2 (Q9FRL8), belonging to the glutathione ascorbate cycle, followed a similar trend. Proteins such as methylglyoxal reductase (Q0PGJ6) and lactoylglutathione lyase (Q9M8R4, O65398, Q04760), as well as L-ascorbate peroxidase (Q1PER6, Q05431), involved in the glutathione ascorbate cycle were also up-accumulated in the last two stages of ripening in MPT compared to YGD, evidencing the activation of methylglyoxal degradation pathways I and III in the MPT cultivar (Figure 4).

### 2.5. DAPs Involved in Carbohydrate Metabolism in the YGD and MPT Cultivars

#### 2.5.1. Glycolysis and Gluconeogenesis

Proteins involved in acetaldehyde synthesis were down-accumulated in the immature and intermediate compared with mature YGD fruits, while proteins involved in gluconeogenesis, and glycan, starch, sucrose and sugar-nucleotide synthesis were slightly up-accumulated in the immature YGD; however, these proteins were down-accumulated in the intermediate and mature fruits compared to immature YGD fruits (Figure 5A). In the case of the MPT cultivar, the proteins involved in gluconeogenesis and the synthesis of glycan, glycogen, starch and sucrose were down-accumulated in the immature stage compared to intermediate and mature fruits; only the proteins involved in the synthesis of acetaldehyde showed a slight up-accumulation in all the analyzed stages of MPT fruits, in comparison with YGD fruits. Proteins involved in gluconeogenesis and the synthesis of glycan, starch, sucrose and sugar-nucleotide synthesis were up-accumulated in the intermediate and mature stages of MPT cultivar (Figure 5A).

The analysis of proteins involved in the gluconeogenesis pathway showed that malate dehydrogenase (P57106, Q9SN86), independent 2,3-bisphosphoglycerate (O04499), phosphoglycerate kinase (Q9LD57, Q9SAJ4), glyceraldehyde 3-phosphate dehydrogenase (Q9FX54, Q5E924), fructose bisphosphate aldolase (Q9ZU52, Q9SJQ9), and glucose 6 phosphate isomerase (P34795, Q8H103) were down-accumulated in the intermediate and mature stage of YGD coconut (Figure 5B). In the case of the MPT cultivar, all these proteins started with low amounts in the immature stage (log2FC −1.07) and increased in the intermediate and mature stages, reaching log2FC +0.36 and +1.07, respectively (Figure 5B). Phosphoglycerate kinase (Q9LD57, Q9SAJ4) and glyceraldehyde 3-phosphate dehydrogenase (Q9FX54, Q5E924) showed the highest accumulation among the proteins involved in the gluconeogenesis pathway in MPT coconut, the former in the intermediate and mature stages and the latter in the intermediate stage (Figure 5B).

The analysis of glycolysis pathway determined the down-accumulation of 6-phosphofructokinase (Q94AA4), diphosphate-fructose-6-phosphate 1-phosphotransferase (Q8W4M5, Q9C9K3), Glucose 6-phosphate isomerase (P34795, Q8H103), triose-phosphate isomerase (Q9SJQ9), glyceraldehyde 3-phosphate dehydrogenase (Q9FX54, Q9LD57, Q9SAJ4, and Q5E924), 2,3-bisphosphoglycerate-independent phosphoglycerate mutase (O04499), phosphoglycerate kinase (Q9LD57, Q9SAJ4, and D7L0K4), enolase (P25696, Q9C9C4), fructose bisphosphate aldolase (Q9ZU52, Q9SJQ9) and pyruvate kinase (Q9FM97, Q9FNN1, Q94KE3, Q9LIK0, and Q9FLW9) in the meat of mature YGD coconut; all proteins showed a decrease in the mature stage in glycolysis I, II and IV (Figure 5C and Appendix A). In contrast, triose-phosphate isomerase (P48491) doubled its amount in the mature stage (log2FC +1.07) (Figure 5B,C and Appendix A).

Regarding glycolysis in the MPT cultivar, most glycolytic enzymes increased as maturity progressed. The protein 6-phosphofructokinase (Q94AA4), increased moderately in the intermediate stage compared to the immature stage, and doubled its amount in the mature stage (Figure 5C and Appendix A). In the case of diphosphate:fructose-6-phosphate 1-phosphotransferase (Q8W4M5, Q9C9K3), involved in glycolysis IV, it started at a low level in the immature stage, but then increased during intermediate and mature stages (Appendix A). Likewise, in YGD, the proteins that increased during the progression of maturity were triose-phosphate isomerase, glyceraldehyde 3-phosphate dehydrogenase, 2,3-bisphosphoglycerate-independent phosphoglycerate mutase, enolase, pyruvate kinase, phosphoglycerate kinase, and the fructose bisphosphate aldolase, all of them observed with low amounts in the immature stage which then increased in the intermediate and mature stages, eventually doubling the amount of initial protein (Figure 5C). In summary, glycolysis I, II and IV were more active in the intermediate and mature stages in the MPT cultivar than in YGD.

#### 2.5.2. The Tricarboxylic Acid Cycle (TCA)

In YGD, proteins involved in the TCA cycle (variation V), i.e., the citrate synthase (P20115), citrate hydro-lyase (cis-aconitate-forming) (Q9SIB9), the cis-aconitate hydratase (Q9SIB9), succinate: quinone oxidoreductase (O82663), and malate dehydrogenase (Q9ZP06, mMDH1, and PMDH1), did not show changes in protein abundance during the maturation of the solid endosperm; the levels were between −0.444 to +0.444 (Figure 6A,B). In contrast, succinyl-CoA synthetase (O82662) decreased from the immature to the mature stage, regardless of whether the TCA type V or II was present. Regarding malate dehydrogenase (P57106, Q9SN86, c-NAD-MDH2, pNAD-MDH), it showed down-accumulation in the intermediate and mature stages in comparison with the immature stage (Figure 6A,B).

When proteins in the TCA cycle were analyzed in MPT endosperm, it was found that citrate synthase, citrate hydro-lyase (cis-aconitate-forming), cis-aconitate hydratase, PMDH1, pNAD-MDH, c-NAD-MDH2 and mMDH1 were initially low in the immature stage, and then increased in the intermediate and mature stages (Figure 6A,B), while the amount of succinate: quinone oxidoreductase (O82663) was stable throughout the maturity process, as was similarly observed in YGD. Some enzymes, e.g., malate dehydrogenase, were only active in the immature stage of YGD. In the case of citrate synthase, it was up-accumulated in the intermediate and mature stages of the endosperm of the MPT cultivar.

### 2.6. DAPs Involved in Lipid Metabolism in the YGD and MPT Cultivars

The proteins involved in fatty acid initiation were up-accumulated in the immature stage of YGD coconut, while they were down-accumulated in intermediate and mature stages; proteins involved in fatty acid elongation and palmitate and stearate synthesis were down-accumulated in the intermediate and mature stages, compared to the immature stage. They were also down-accumulated in the mature stage compared to the intermediate stage (Figure 7A). In the MPT cultivar, the proteins involved in fatty acid initiation showed constant accumulation, remaining at the basal level. Proteins involved in fatty acid elongation and palmitate and stearate synthesis were up-accumulated in the intermediate and mature stages compared to the immature stage (Figure 7A).

The analysis of palmitate biosynthesis showed that 3-oxoacyl-[ACP] reductase (P33207) and 2,3,4-saturated fatty acyl-[ACP]: NAD+ oxidoreductase (Q9SLA8) were down-accumulated in the YGD, while the 3-hydroxyacyl-[ACP] dehydratase (Q9LX13) was up-accumulated, starting from log2FC −1.33 in the immature stage and increasing to log2FC +1.33 in the mature stage (Figure 7B). Regarding the MPT coconut, 3-hydroxyacyl-[ACP] dehydratase and 2,3,4-saturated fatty acyl-[ACP]NAD+ oxidoreductase (Q9SLA8) showed an up-accumulation trend during the maturity stages, with a log2FC from −0.444 to +4 (Figure 7B). The analysis of the protein dynamics involved in fatty acid elongation-saturation showed that three enzymes were down-accumulated in the mature stage of YGD, while the same enzymes were moderately up-accumulated in the mature stage of the MPT coconut (Figure 7C).

### 2.7. Other Proteins

Some proteins were shared between the two cultivars and the three maturation stages showing similar behaviors, such as the embryonic proteins DC-8 (A0A8B8ZT31), vicilin-like antimicrobial peptides 2-2 (A0A8B7BEE0), vicilin-like seed storage protein (A0A8B7BPL4), elongation factor 2 (AT1G56070), tubulin alpha chain (AT4G14960), tubulin beta-chain (AT5G12250), heat shock protein 83 (AT5G52640), and late embryogenesis abundant protein of 31-like (A0A8B7CDB9), whose accumulation increased throughout maturation of both cultivars (Appendix A).

The exclusive DAPs found in the YGD cultivar included a putative mitochondrial cytochrome c1-1 heme protein (AT5G40810), the ribonuclease TUDOR 1 (AT5G07350), the coatomer subunit alpha-1 (AT1G62020), the ERBB-3 binding protein 1 (AT3G51800), the adenylate kinase 4 (AT5G50370), and the hypothetical protein COCNU_10G003830 (AT3G20050), which were down-accumulated along the maturation event (Appendix A). In contrast, DAPs, such as actin-101 (AT5G09810), putative mitochondrial heat shock 70 kDa (AT5G09590), proteasome subunit alpha type-5 (AT3G14290), putative mitochondrial formate dehydrogenase (AT5G14780), aminotransferase ALD1 (AT4G33680), serine/threonine-protein kinase SAPK7 (AT1G10940), and glutathione S-transferase 3 (A0A8B7C1W8), increased in the intermediate stage but decreased in the mature stage (Appendix A).

In the MPT coconut, DAPs included polyadenylate-binding protein 5 (A0A8B7MTK7), acetylornithine deacetylase (AT4G17830), putative isopentenyl-diphosphate delta-isomerase I (AT5G16440), mitochondrial gamma-aminobutyrate transaminase 1 (AT3G22200), actin-101 (AT5G09810), two-component response regulator 24 (A0A8B7D411), which were down-accumulated throughout ripening (Appendix A). On the contrary, DAPs such as hypothetical protein COCNU_08G003720 (AT3G01650), putative low-temperature-induced 65 kDa protein (A0A8B7C9J1), putative coatomer beta’-1 (AT1G79990), FT-interacting protein 7 (AT3G57880), putative seed maturation protein (AT3G10960), monocopper oxidase-like protein SKU5 (AT4G12420), and cinnamoyl-CoA reductase 1 (AT2G33590) were up-accumulated throughout the maturation process (Appendix A).

## 3. Discussion

Metabolic changes linked to physiological, biochemical and molecular processes associated with seed and fruit maturity are influenced by endogenous and external signals [20]. Changes include modifications of the cell wall, the accumulation and degradation of starch and triacylglycerol (TAG), among others [21]. On the other hand, color, hardness/softening and sugar content are used to monitor fruit development and maturity [22]. Most studies classify the maturity of the fruits into three main stages: unripe, breaker and ripe, as in *Persea americana* Mill. [23], or endosperm development, embryo development, seed dehydration and maturation like in *Brassica napus* L. [24].

Coconut seed ripening occurs in three stages: immature, intermediate, and mature [10]. However, the maturation of the coconut fruit does not follow the same pattern as other fruits; coconut meat begins to develop at the opposite end to the stalk and the embryo; it grows during maturation, and the amount of different proteins in the solid endosperm increases [25]. In the immature stage, the endosperm of the coconut has a soft consistency and hardens as it matures, and the fruit loses water. In this report, coconut cultivars YGD and MPT were compared; the MPT coconut was heavier and contained a thicker solid endosperm than YGD (Figure 1 and Appendix A). According to Farooq et al. [26], the development of the endosperm requires large amounts of energy, which is fulfilled mainly by glycolysis.

The regulation of fatty acid and carbohydrate metabolisms in the developing endosperm of oil seeds is less understood than in legumes [13,27]. In this work, an enrichment analysis of biological processes identified proteins associated with the response to stress, particularly to heat and viruses, as the most represented core proteome in the YGD and MPT cultivars (Figure 2A–C). The identification of stress response proteins among enriched biological processes is consistent with the fact that coconut palm grows near coastal areas, where solar radiation and heat are high [28].

Proteomics showed metabolic differences between both cultivars (Figure 2B,C). Regarding carbohydrate metabolism (photosynthesis and glycolytic processes), the coconut is a C3 plant, i.e., initial CO_2_ fixation is in ribulose 1,5-bisphosphate that immediately breaks down into 1,3-bisphophoglycerate. After three rounds of CO_2_ fixation, the first 3-phosphoglycerate is biosynthesized and channeled to produce sucrose or starch, or to triose catabolism to produce TCA cycle intermediaries and NADH, FADH2 and ATP for cell redox balance and energy homeostasis [29]. Glycolysis was higher in MPT than in YGD (Figure 2B,C), and the former accumulated greater amounts of lipids in the solid endosperm. This was expected since lipid biosynthesis requires carbon and energy sources, which are provided by glycolysis and the TCA cycle [30].

Increased glycolytic flux results in the accumulation of methylglyoxal, a toxic byproduct of glycolysis [31]. This explains why lactoylglutathione lyase and methylglyoxal reductase, proteins involved in methylglyoxal detoxification, were over-accumulated as ripening progressed in the MPT coconut; meanwhile, these enzymes were stably produced in YGD (Figure 4). The accumulation of methylglyoxal is evident in many plants with intracellular oxidative stress [32,33]. Similarly, the proteins involved in coping with oxidative stress were also up-accumulated with the progress of ripening in MPT. Superoxide dismutase, which detoxifies superoxide and catalase, an efficient hydrogen peroxide scavenger, was up-accumulated in MPT coconut.

The PCA evidently grouped the replicates of immature, intermediate and mature fruits from each coconut cultivar, which supported the reproducibility of the sample collection, processing and analysis (Figure 3A,E). The volcano plots allowed for an analysis of the fold change vs. the significance of protein up-accumulation in the Int/Imm, Mat/Imm and Mat/Int in the YGD and MPT cultivars. The ALATA 2 protein, detected in the Int/Imm of the YGD, is involved in the transamination reactions between L-alanine and 2-oxoglutarate, and the reverse reaction between L-glutamate and pyruvate, an essential intermediate for the metabolism of carbohydrates and lipids. On the other hand, the UDP-glucose 4 epimerase (GEPI48) catalyzes the interconversion of UDP-galactose (UDP-Gal) to UDP-glucose (UDP-Glc), thus channeling glucose towards more complex carbohydrate biosynthesis or carbohydrate catabolism to produce pyruvate [34,35].

In the intermediate stage of YGD, there were down-accumulated proteins involved in chaperone refolding activity (FAM10), as well as annexin D1 (ANXD1), a membrane-bound protein involved in the reduction in oxidative stress [36] (Figure 3B), suggesting that the transition from the immature to intermediate stage in this cultivar is a step with low proteolytic activity and oxidative stress. Likewise, in the mature stage compared to the immature stage, TCPQ, a subunit of chaperons that assists in the correct folding of nascent proteins [37], and HSP904, another molecular chaperone involved in the response to various stresses [38], were down-accumulated (Figure 3C), as well as the BCCP complex protein that regulates the first step in fatty acid biosynthesis [39], and proteins involved in fatty acid synthesis initiation and fatty acid elongation saturation (Figure 7A). Therefore, the transition to the mature stage in YGD occurred with low oxidative stress and, more importantly, it was strongly associated with a decrease in proteins involved in lipid biosynthesis (Figure 3C). In the mature stage, compared to the intermediate stage in cultivar YGD, maturity was also accompanied by a decrease in proteins related to carbohydrate metabolism, for example, phospho-fructokinase-6 (PFK6) and dihydrolipoyl dehydrogenase 2 (LPD2) (Figure 3D), in agreement with the decrease in proteins involved in the synthesis of glycan gluconeogenesis, sucrose and starch, among others, observed in our study (Figure 5A).

In the case of the MPT cultivar, the transition from the immature to intermediate stage (Figure 3F,G) was associated with an accumulation of proteins involved in development, for example, the calcium-binding EF-hand protein family (CbEF), a biological switch that regulates signal transduction by Ca^2+^ and controls cell growth and plant development [40]. Similarly, there was an accumulation of proteases, like carboxypeptidases, that supply nitrogen. Other up-accumulated proteins included glucan endo-1,3-β-glucosidase (GbG), which is responsible for β-glucan cleavage and releases glucan oligosaccharides, and the early nodulin-like protein 1 (Enod1), which transports glucose, fructose or sucrose for plant development [41] (Figure 3F). These proteins increased throughout maturation (Figure 3G,H), consistent with the cell expansion and development of the seed and fruit.

The proteins most down-accumulated in the intermediate stage compared to the immature stage in MPT were UDP-glucose 6-dehydrogenase 3 (UGDH3) and sucrose synthase 4 (SUS4) (Figure 3F), with the former being involved in the synthesis of hemicellulose for the plant cell wall [42] and the latter catalyzing the reversible cleavage of sucrose into fructose and UDP-glucose or ADP-glucose, which could then be channeled into energy production or for cell wall synthesis [43]. In MPT, the differentiation from the immature to intermediate solid endosperm was accompanied by a decrease in hemicellulose biosynthesis, possibly preventing the reinforcement of the cell wall, therefore facilitating cell expansion.

In the mature stage, compared to the immature stage in MPT, the down-accumulated proteins included a biotin carboxylase (BC) enzyme that carboxylates the N1′ of biotin [44]; glycerol-3-phosphate dehydrogenase (SDP6) that reduces dihydroxyacetone phosphate to glycerol-3-phosphate, a key step in glycerolipid metabolism [45]; and the phospho-fructokinase-6 (PFK6), a key enzyme in the carbohydrate metabolism [46] (Figure 3G). This is consistent with the decrease in proteins involved in the biosynthesis of carbohydrates (Figure 5A) and fatty acids (Figure 7A) observed in the mature MPT.

Comparing the proteomics of the mature stage and the intermediate stage in MPT, a down-accumulation of aldehyde dehydrogenase (ALDH) was observed in the mature stage (Figure 3H). ALDH has been described in the biosynthesis of aromatic volatiles during maturation in postharvest banana [47] and in the degradation of fatty acids [48]; its down-accumulation in the MPT coconut is understood since solid endosperm from the mature fruits of MPT is used for copra production due to its high fatty acid content.

Regarding APX2, it detoxifies the H_2_O_2_ during biotic and abiotic stresses [49]. APX2 was down-accumulated in the mature stage, while glutathione S-transferase (GST) was up-accumulated (Figure 3H), showing that antioxidant proteins are well coordinated in the MPT endosperm (Figure 4) to cope with different kinds of stresses. A primary supply of myo-inositol is more necessary in the intermediate stage compared to the mature stage, since the INO2 protein was down-accumulated in the latter stage. This coincides with reports in *Arabidopsis thaliana* and *Glycine max*, where the INO2 enzyme is necessary for the normal development of endosperm and embryo [50].

YGD and MPT coconut cultivars also showed differences in gluconeogenesis (Figure 5B). During gluconeogenesis and the Calvin–Benson cycle, the photosynthetic activities of the phosphoglycerate kinase isoenzyme 1 and 3 were down-accumulated in the YGD cultivar; the decrease in photosynthetic activity and glycolysis may be a strategy for metabolic adjustment to optimize plant or fruit growth [51]. Yang et al. [52] reported that most DAPs are associated with carbohydrate metabolism in other varieties of coconut in response to low temperature. In *Camellia oleifera*, Ye et al. [48] found that the metabolism of carbohydrates and lipids was among the DAPs, but each metabolism integrated the results from all stages, and, thus, the dynamics of DAPs between these two plants cannot be compared stage by stage.

Enolase catalyzes the conversion of 2-phosphoglycerate to phosphoenolpyruvate (PEP), an important step in all classes of glycolysis (I, II, and IV), (Figure 5C and Appendix A). Hassan et al. [53] found that enolase was up-accumulated during the development of the mesocarp of oil palm, related to the production of oil in the mesocarp. In other fruits, such as Chinese bayberry [54] and apricot [55], the production of enolase protein also increased throughout ripening. In our work, it was found that enolase protein was more accumulated in MPT than in YGD during seed maturation. Again, this is in agreement with the use of MPT solid endosperm for oil production. As a product of glycolysis, pyruvate is converted to acetyl-CoA by the pyruvate dehydrogenase (PDH) complex, and acetyl-CoA has several fates: as a substrate for the glyoxylate cycle, the TCA cycle, or a precursor for fatty acid elongation and lipid biosynthesis [15,56]. Glycolysis was more active in MPT than in YGD (Figure 5C), and pyruvate may be channeled to support lipid biosynthesis in this case.

Regarding malate dehydrogenase (MDH), a key enzyme involved in carbohydrate and lipid metabolisms, it was found to be up-accumulated in the TCA cycle in the MPT cultivar (Figure 6), similar to seeds of castor bean, where glycolysis and TCA were among the most active metabolic processes [57]. Likewise, in *Brassica napus*, MDH was up-accumulated in the cytosol and mitochondria of seeds having a higher content of oil [58], supporting a key role of MDH in oilseeds.

In plants, the synthesis of fatty acids occurs in the plastids while their breakdown occurs in the peroxisomes, the glyoxysomes, and to a lesser degree in the mitochondria, and elongation occurs in the endoplasmic reticulum [59,60]. However, although the compartment regulation of plant lipid biosynthesis is clear, the temporal and spatial regulation diverge among plants, and oil content in seeds greatly varies in quantitative and qualitative terms. Studies show that lauric acid is 65% in *C. nucifera* and it is 85% in *E. guineensis* [61,62], while oleic acid predominates in *Arachis hypogaea* (45–56%), *P. americana* (60%), and *Styrax tonkinensis* (60%) [63,64,65]. According to the previous findings, it is necessary to further understand the metabolic pathways of carbohydrates and lipids of oilseeds such as coconut, since there may be differences among the plant families regarding the regulation of these metabolic pathways [15]. In the case of the coconut tree, Maskromo et al. [66] and Tenda et al. [11] described significant differences in the fatty acid content of tall and dwarf varieties, which allows us to hypothesize that there is a correlation between the fatty acid content and the accumulation of proteins related to that metabolism.

There was a higher accumulation of proteins involved in the biosynthesis of fatty acids in the intermediate and mature stages of the MPT cultivar than in YGD (Figure 7A), suggesting a high production of acetyl-CoA. These results are in concordance with coconut seed uses: MPT for copra and oil extraction and YGD for water consumption (Prades et al., 2012). Acetyl-CoA carboxylase (ACCase) regulates the most essential step in fatty acid biosynthesis by converting acetyl CoA to malonyl CoA [67]. In oilseeds, acetyl-CoA combines with oxaloacetate in the glyoxylate cycle; then, succinate, one of the products of this cycle, enters mitochondrial TCA and gluconeogenesis, where it is a precursor to hexoses, necessary for postgermination growth [68]. Likewise, the enzyme 3-hydroxyacyl-ACP dehydratase, which transforms 3-hydroxyacyl-ACP to enoyl-ACP and enoyl-ACP reductase to acyl-ACP in palmitate biosynthesis and fatty acid elongation, showed the greatest up-accumulation in the MPT cultivar (Figure 7B,C), similar to that reported in *B. napus*, *R. communis*, *P. americana*, and the mesocarp of *E. guineensis* [69].

It can be said that during the development and germination of the seed, the proteome is greatly represented by proteins with enzymatic activities, such as those involved in the metabolism of lipids and carbohydrates, while the proteome associated with structural or storage functions is represented by globulins, vicilin, and oleosins, among others, which play a crucial role in supporting cell shape and structure or as energy reserves for the development of embryos and seedlings [70].

In our work, during the protein profiling of coconut solid endosperm at different stages of development of YGD and MPT, it was observed that vicilin, glutelin type A-1, 11S globulin, cocosin 1, late embryogenesis abundant protein 3 (LEA), and heat shock 70 kDa proteins were detected in the “core proteome” (Appendix A). Our results were in line with those of Garcia et al. [71], who reported 11S globulin and 7S vicilin in the solid coconut endosperm; the former, called cocosin, is known as the main coconut seed storage protein. Similarly, Patil and Benjakul [72] reported glutelin, and Dave [73] found oleosins involved in the stabilization of oil bodies in the solid endosperm. Our results evidenced that coconut storage proteins and heat shock proteins were distributed from the immature to mature stages in YGD and MPT cultivars, suggesting that their presence is crucial for endosperm development, accumulation and structural formation. Biochemical evidence in *Arabidopsis* suggests that storage proteins contribute to seed germination vigor, support early seedling growth, and could have a role in seed longevity [74]. Regarding the heat-shock proteins, these are important in fruit maturation, protein stabilization, protein refolding, and protection against environmental stresses [30,33].

## 4. Materials and Methods

### 4.1. Plant Material Collection

Yucatan green dwarf and Mexican pacific tall coconuts were harvested from a coconut plantation located at 20°24′03.95″ N and 89°30′56.86″ W in San Crisanto, Yucatan, Mexico. The youngest inflorescence is at the top of the palm, closest to the apical meristem, and shows a closed spathe (0 stage, not one ovule fertilized in the ovaries). Coconut palms produce one inflorescence per month, with the first fertilized ovules occurring in inflorescence at stage 1, so from inflorescence stage 0, the age of inflorescences and fruits increases each month in the direction to the ground [75,76]. The stages of fruit maturity were classified as immature for fruits from 6 to 8 months, intermediate for fruits from 9 to 10 months, and mature for fruits from 11 to 14 months. The fruits were disinfested with sterile distilled water and manually dehusked, the endocarp was cut with a sterile steel knife, and 20 g of meat was stored at −80 °C until further processing.

### 4.2. Proteome Extraction from Coconut Solid Endosperm

Proteins were extracted according to Islas-Flores et al. [77], with modifications. Briefly, one gram (1 g) of solid endosperm was ground to a fine powder with a sterile mortar and pestle in liquid nitrogen, and then 1 mL of buffer was added, which contained 50 mM Tris-HCl, pH 7.4, 50 mM NaCl, 10% glycerol, 1 mM EDTA, 1 mM EGTA, 5 mM ß-mercaptoethanol, 1 mM phenylmethylsulfonyl fluoride (PMSF), and ¼ of pill protease inhibitor cocktail (Roche, Basel, Switzerland). After homogenization, it was centrifuged for 30 min at 13,000 rpm and 4 °C, and the supernatant was stored at −20 °C until use. Each sample from each developmental stage was extracted individually; biological replicates were performed in triplicate.

### 4.3. Protein Quantification and Analysis by SDS-PAGE

Protein concentration in each supernatant was determined according to the Bradford method [78]. Protein analysis (30 µg) was performed on 12% SDS-PAGE gel and electrophoresed at 80 V for 2 h at room temperature. After electrophoresis, the gels were incubated overnight in Coomassie blue stain and distained in a solution containing 10% acetic acid, 25% methanol, and 65% sterile distilled water.

### 4.4. Sample Preparation for Proteomics

For each sample, 1200 µg of proteins was precipitated with chloroform–methanol according to Burgos-Canul et al. [79]. The proteins (100 µg per sample) were reduced for 45 min at 60 °C with 10 mM Tris (2-carboxyethyl) phosphine (TCEP), alkylated with 30 mM iodoacetamide (IA) for 1 h at room temperature in the dark, then quenched with 30 mM dithiothreitol (DTT). The proteins were then precipitated overnight at −20 °C in cold acetone and centrifuged at 10,000× *g* for 15 min at 4 °C. The pellets were dried for 15 min at room temperature, and the proteins were suspended in 100 µL of digestion buffer containing 50 mM triethylammonium bicarbonate (TEAM) and 0.1% sodium dodecyl sulfate (SDS) and digested overnight at 37 °C with trypsin 1:30 (*w*/*w*) (Promega, Madison, WI, USA, Trypsin Gold, Mass Spectrometry Grade). After this period, freshly prepared trypsin was added 1:60 w/w and incubation occurred for 4 h at 37 °C. The digestion reaction was stopped at 80 °C [80]. The trypsin-digested peptides were desalted and dried at room temperature.

### 4.5. TMT Labeling and Peptide Fractionation

Tandem mass tags (TMT), in this case TMT-9plex reagent, were added with acetonitrile and gently agitated. Peptide samples were resuspended in 100 mM triethylammonium bicarbonate (TEAB) buffer for 5 min and conjugated with TMT at room temperature for 1 h, following the manufacturer’s instructions. The Yucatan green dwarf (YGD) and Mexican pacific tall (MPT) samples were labeled and analyzed separately. For each cultivar, the peptides of coconut in the immature stage were labeled with the reagents 126, 127N, and 128N; the intermediate stage proteins were labeled with 128C, 129N, and 129C; and the mature stage proteins were labeled with 130N, 130C, and 131. Peptide labeling was stopped with 5% hydroxylamine, incubating at room temperature for 15 min. All samples from the same variety were pooled and fractionated using high-pH reversed-phase C18 cartridges (Thermo Scientific, San Jose, CA, USA). The pooled samples were desalted with C18 cartridges, cleaned with ZipTip, and dried using a CentriVap (Labconco, Kansas City, MO, USA).

### 4.6. Nano-LC-MS/MS and Synchronous Precursor Selection (SPS)-MS3

The dried samples were dissolved with 0.1% formic acid in LC-MS-grade water (solvent A) and 20 µL of this solution was injected onto a nano-LC platform (UltiMate 3000 RSLC system, Dionex, Sunnyvale, CA, USA) through a nanoviper C18 trap column (3 µm, 75 µm × 2 cm, Dionex), and fractionated on an EASY spray C-18 RSLC column (2 µm, 75 µm × 25 cm) at a flow rate of 300 nL/min.

The mobile-phase gradient was as follows: a 10 min gradient was set using Solvent A and 0.1% formic acid in 90% acetonitrile (Solvent B), 10 min Solvent A, 7–20% Solvent B for 25 min, 20% Solvent B for 15 min, 20–25% Solvent B for 15 min, 25–95% Solvent B for 20 min, and 8 min Solvent A. The nano-LC was hyphenated to an Orbitrap Fusion Tribrid (Thermo-Fisher Scientific, San Jose, CA, USA) mass spectrometer equipped with an “EASY Spray” nano-ion source (Thermo-Fisher Scientific, San Jose, CA, USA). The mass spectrometer was operated in positive ionization mode with the nanospray voltage set at 3.5 kV and a source temperature of 280 °C. Caffeine, Met-Arg-Phe-Ala (MRFA) and Ultramark 1621 were used for external calibration.

Full MS scans were performed on the Orbitrap analyzer at a resolution of 120,000 (FWHM), scan range of 350–1500 *m*/*z*, AGC of 2.0 × 10^5^, maximum injection time of 50 ms, intensity threshold of 5.0e3, dynamic exclusion 1 at 70 s, and mass tolerance of 10 ppm. For the MS2 analysis, the 20 most abundant MS1s were isolated with charge states set to 27. The fragmentation parameters included CID with 35% collision energy and activation Q of 0.25, an AGC of 1.0 × 10^4^, a maximum injection time of 50 ms, a precursor selection mass range of 400–1200 *m*/*z*, a precursor ion exclusion width low of 18 *m*/*z* and high of 5 *m*/*z*, isobaric tag loss TMT and in-trap detection ion; MS3 spectra were acquired using a 10-notch SPS isolation. MS3 precursors were fragmented using HCD with 65% collision energy and analyzed using the Orbitrap at 60,000 resolution with a 120–500 *m*/*z* scan range, 2 *m*/*z* isolation window, 1.0 × 10^5^ AGC and maximum injection time of 120 ms with 1 microscan.

The spectra acquired using MS/MS and (SPS)-MS3 were analyzed with the program Proteome Discoverer 2.4 (Thermo-Fisher Scientific Inc.). Data were processed using the search engines Amanda and Mascot server (version 2.4.1, Matrix Science, Boston, MA, USA) and the Sequest HT algorithm, in which searches were conducted against *Arabidopsis* and the sequences of those of the Arecaceae family downloaded from the NCBI protein database. The sequences were concatenated into one single file and were filtered to remove the undetermined amino acids. To reduce the protein redundancy, sequences were collapsed with the CDHIT [81] program using 0.98, 5, and 0 for -c, -n, and -t parameters, respectively. Full-tryptic protease specificity, two missed cleavages, carbamidomethylation of cysteine (+57.021 Da) and TMT 9-plex N-terminal/lysine residues (+229.163 Da) were considered. Methionine oxidation (+15.995 Da) and deamidation in asparagine/glutamine (+0.984 Da) as dynamic modifications were considered as well.

Protein identification was carried out at a lower resolution in the linear ion trap with tolerances of ±10 ppm and ±0.6 Da. Peptide hits were filtered for a maximum of 1% false discovery rate (FDR) using the Percolator algorithm [82]. In addition, TMT quantification was performed at the MS3 level in the Orbitrap analyzer with the precursor co-isolation filter set to 45%. Data are available via ProteomeXchange with identifier PXD036949 (https://www.ebi.ac.uk/pride/, accessed on 22 September 2022).

### 4.7. Bioinformatics Analyses

Protein and peptide abundance were generated on the Proteome Discoverer 2.4 platform. Normalization was carried out based on the total peptide amount. The fold change in protein abundance was determined in the ratio Int/Imm, Mat/Imm, and Mat/Int. Differential proteins (fold change >1.5 and <0.66 (log2 fold change >0.59 and <−0.59)) were analyzed based on biological process gene ontology (GO) enrichment using the platform DAVID Bioinformatics Resources 6.8 (https://david.ncifcrf.gov/summary.jsp, accessed on 22 September 2022), and the resulting GO terms were then summarized by finding a representative subset with a simple clustering algorithm that relies on semantic similarity measures using the REVIGO software (http://revigo.irb.hr, accessed on 22 September 2022). We used protein homologues to *Arabidopsis*, *E. guineensis,* and *P. dactylifera* as reference databases. The output data were presented using Treemaps, displaying a two-level hierarchy of GO terms—each rectangle is a single cluster representative, and the representatives were joined into “superclusters” of loosely related terms. PCA-based normalized protein abundances were calculated to visualize the differences in protein abundances between conditions. Log2 ratios Int/Imm, Mat/Imm, and Mat/Int; significant values (*p*-value −log10), and a Manhattan distance as a criterion for ranking hits in the platform VolcaNoseR (https://huygens.science.uva.nl/VolcaNoseR2/, accessed on 22 September 2022) were calculated. The plant metabolic database (https://plantcyc.org/, accessed on 22 September 2022) was used and samples were computed using the cell overview/omics viewer tool to generate pathway diagrams and protein accumulation intensities based on the Int/Imm, Mat/Imm, and Mat/Int ratios.

## 5. Conclusions

The present work adds new information to the proteomic landscape associated with coconut solid endosperm, focusing on the metabolisms of carbohydrates and lipids in the immature, intermediate, and mature stages in two contrasting coconut cultivars, MPT and YGD. The former is a high lipid and fatty acid producer, while the latter is a less efficient producer of lipids but is highly valued as young (immature) fruits because of the taste of its coconut water. Interestingly, proteins involved in carbohydrate metabolism, e.g., in glycolysis, were more greatly accumulated in the mature stage of MPT than in YGD; this most likely occurs in order to synthesize precursors for fatty acids through enzymes such as MDH and enolase, whose products may be channeled for fatty acid production. Carbohydrate metabolism was higher in the immature stage of YGD, but was reduced in the intermediate and mature stages. Future studies combining proteomics analysis with metabolomics may help to elucidate the fate of those carbohydrates during the intermediate and mature stages of solid endosperm development. In agreement with our hypothesis, the proteins of solid endosperm involved in fatty acid metabolism were most accumulated in the intermediate and mature stages of MPT compared to YGD.

By investigating dynamic proteomics in carbohydrate and lipid metabolisms during solid endosperm maturation, our results, for first time, provide findings that contribute to a better understanding of the biochemical events related to maturation of the coconut. Although proteomics revealed differences in carbohydrate and lipid metabolism between MPT and YGD, the regulation of these metabolisms was found to be conserved with other fruits, suggesting that the regulation of carbohydrate and lipid metabolisms is ancient and preserved, regardless of the taxonomy or physiology of the fruits. Further analyses, for example focusing on the coconut endocarp, mesocarp, and pericarp, can help to unravel metabolic differences between coconut and other fruits, e.g., climacteric fruits. The present work greatly expands the current knowledge surrounding the fundamental biology of this non-climacteric species, which, in general terms, is a group still poorly studied.

## Figures and Tables

**Figure 1 ijms-24-10431-f001:**
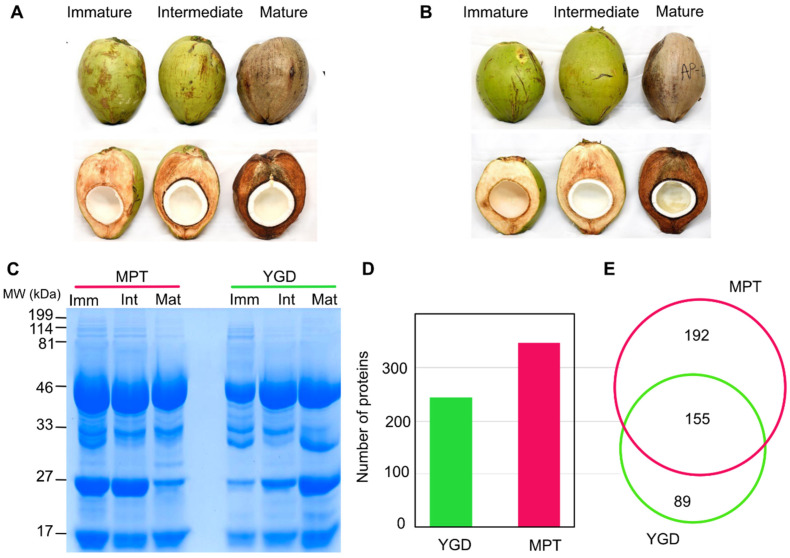
Coconut ripening and proteins identified in Yucatan green dwarf (YGD) and Mexican pacific tall (MPT) cultivars. Visual changes in fruit ripening of YGD (**A**) and MTP (**B**). SDS-PAGE protein profile of solid endosperm during ripening (**C**), the protein molecular weight standard is indicated on the left side of the figure. Number of proteins identified in YGD and MTP by TMT-SPS-MS3 (**D**). Proportional Venn diagram representation of proteins identified in each cultivar (**E**).

**Figure 2 ijms-24-10431-f002:**
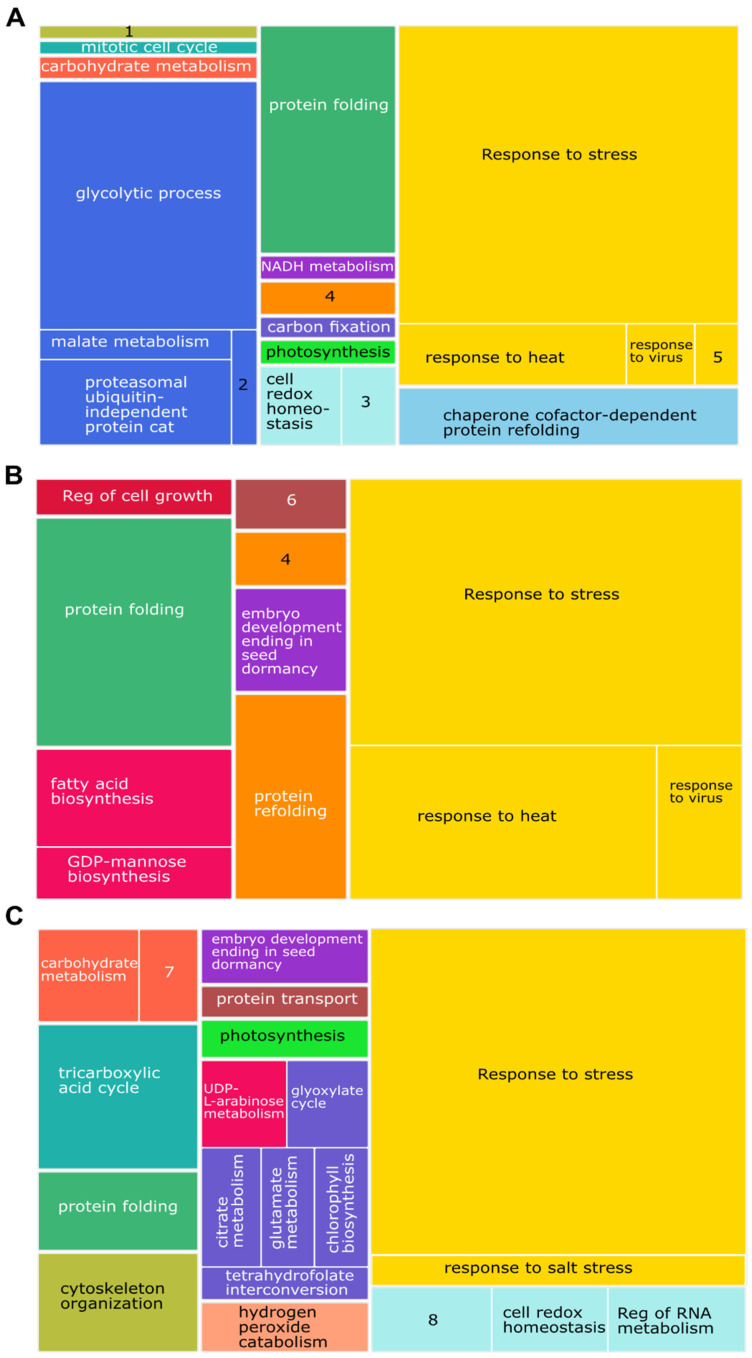
Enrichment of the biological processes found in solid endosperm of coconut visualized as a TreeMap. Proteins found in both cultivars (core proteome, **A**). Proteins identified particularly in YGD (**B**) and MPT (**C**) cultivars. Each box represents the *p*-value weight within the treemap. Components (boxes) of the same color represent semantic similarity (medium allowed similarity, 0.7). The most enriched biological processes are shown as larger components within the map. 1, microtubule cytoskeleton org; 2, Fructose 1,6-bisphosphate metabolisms; 3, regulation of vacuole fusion, non-autophagic; 4, microtubule-based process; 5, response to arsenic-containing substance; 6, retrograde vesicle-mediated transport, Golgi to endoplasmic reticulum; 7, carbohydrate derivative metabolism; 8, negative regulation of response to water deprivation.

**Figure 3 ijms-24-10431-f003:**
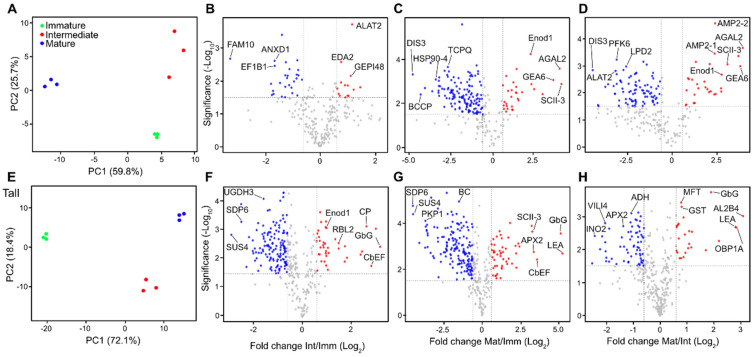
Comparative proteomics analyses of solid endosperm during ripening. Principal component analysis (PCA) of protein abundances of YGD (**A**). Significant differential accumulated proteins represented in volcano plots in the Log 2 ratios intermediate/immature (**B**), mature/immature (**C**) and mature/intermediate samples (**D**). PCA of protein abundances of MPT (**E**). Significant differential accumulated proteins are represented in (**F**–**H**). The *x*-axis corresponds to the log2 fold change value, and the *y*-axis displays the −log10 *p*-value. The points in blue represent the down-accumulated proteins (<1/1.5-fold change, FC), those in red represent the over-accumulated proteins (≥1.5 FC) and those in grey represent proteins with not changes in log2 fold change. Alanine aminotransferase 2 (ALAT2), L-ascorbate peroxidase 2 (APX2), Alpha-galactosidase 2 (AGAL2), Annexin D1 (ANXD1), Aldehyde dehydrogenase family 2 member B4 (AL2B4), Biotin carboxylase (BC), Calcium-binding EF-hand family protein (CbEF), Carboxypeptidase (CP), Elongation factor 1-beta 1 (EF1B1), putative Elongation factor 1-alpha (EF1A4), early nodulin-like protein 1 (Enod1), Em-like protein (GEA6), Dihydrolipoyl dehydrogenase 2 (LPD2), Inositol-3-phosphate synthase isozyme 2 (INO2), Probable fructokinase-6 (PFK6), Glycerol-3-phosphate dehydrogenase (SDP6), Heat shock protein 90-4 (HSP90-4), Glutathione S-transferase family protein (GST), putative DIS3-like exonuclease 2, UDP-glucose 4-epimerase (GEPI48), putative FAM10 family protein, glucan endo-1,3-beta-glucosidase-like (GbG), 11 kDa late embryogenesis abundant protein (LEA), RHOMBOID-like protein 2 (RBL2), 11 kDa late embryogenesis abundant protein (LEA), putative DIS3-like exonuclease 2, Protein MOTHER of FT and TFL1 (MFT), Oil-body-associated protein 1A (OBP1A), T-complex protein 1 subunit theta (TCPQ), Plastidial pyruvate kinase 2 (PKP1), Plastidial pyruvate kinase 2 (PKP1), Sucrose synthase 4 (SUS4), serine carboxypeptidase II-3 (SCII-3), putative serine protease (EDA2), UDP-glucose 6-dehydrogenase 3 (UGDH3), vicilin-like antimicrobial peptides 2-2 (AMP2-2), and Villin-4 (VILI4).

**Figure 4 ijms-24-10431-f004:**
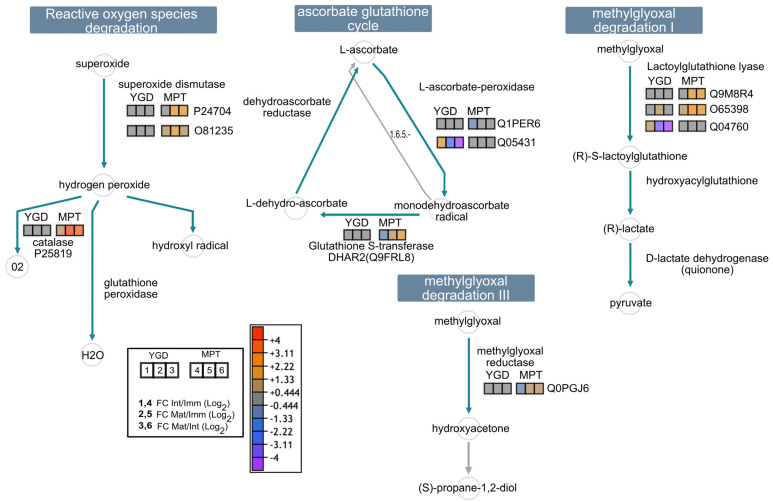
Visual representation of differential proteins associated with enzymes related to reactive oxygen species (ROS) and methylglyoxal detoxification pathways in YGD and MPT cultivars. Pathway reconstruction was carried out using the “cellular overview” tool on the Plant Metabolic Network (PMN) database platform. Numbers in squares specify, 1,4: log2 FC intermediate/immature (Int/Imm); 2,5: log2 FC mature/immature (Mat/Imm); 3,6: log2 FC mature/intermediate (Mat/Int). Numbers 1,2,3 indicate YGD variety, while 4,5,6 indicate MPT variety. The same applies to the following figures.

**Figure 5 ijms-24-10431-f005:**
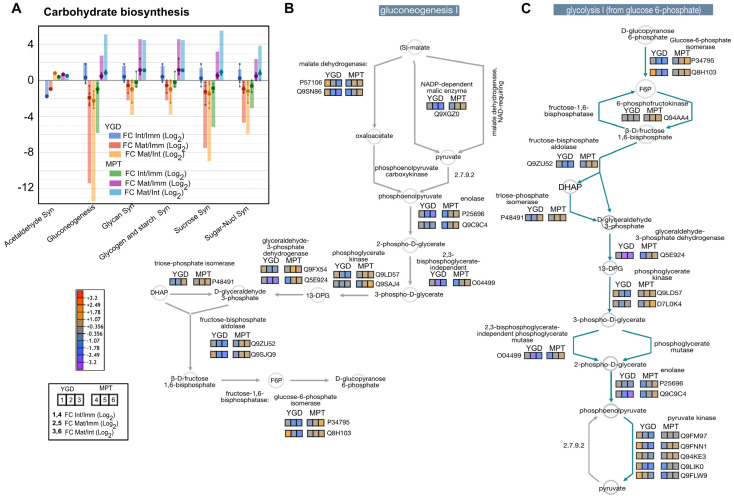
Visual representation of differential proteins associated with the carbohydrate biosynthetic main pathways. Pattern of accumulation of carbohydrate-related proteins determined by TMT-SPS-MS3. Small circles specify the log2FC values of each protein; the big circles are the average of the log2FC values, and columns indicate the sum of the log2FC (**A**). Dynamic of identified differential enzymes in the gluconeogenesis I (**B**) and glycolysis I (from glucose 6-phosphate) (**C**). Glycolysis II and IV are presented in Appendix A. The pathway reconstruction was carried out by using the “cellular overview” tool in the Plant Metabolic Network (PMN) database platform.

**Figure 6 ijms-24-10431-f006:**
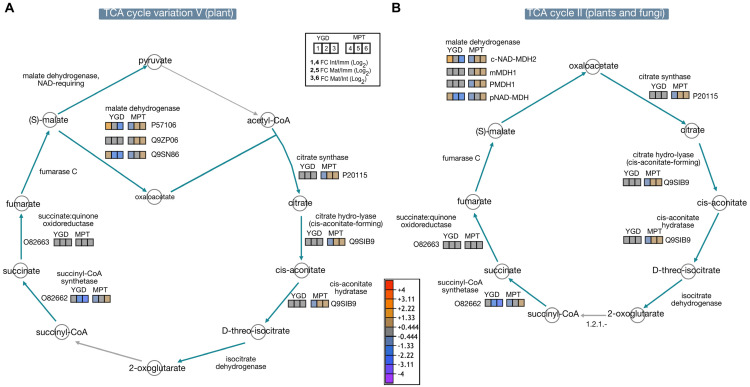
Visual representation of differential proteins associated with TCA cycle. Dynamic of identified differential enzymes in the TCA cycle V (plants, **A**) and II (plants and fungi, **B**).

**Figure 7 ijms-24-10431-f007:**
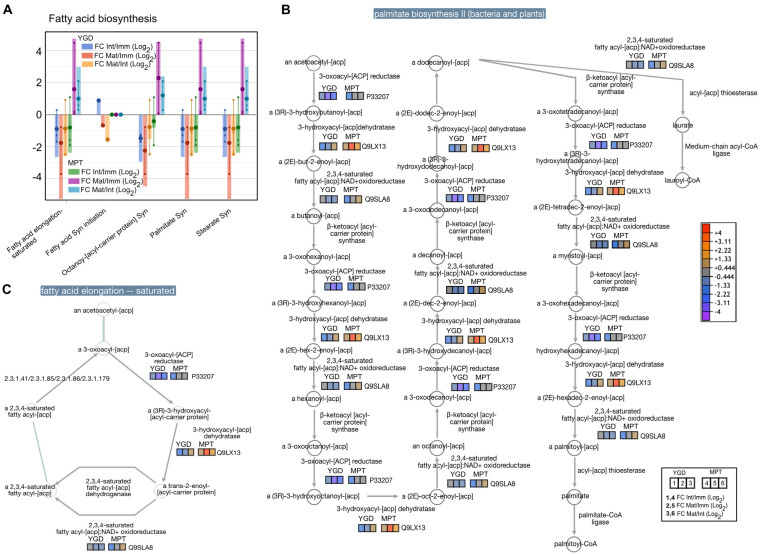
Visual representation of differential proteins associated with the fatty acid biosynthetic pathways. Pattern of accumulation of lipid-related proteins determined using TMT-SPS-MS3. Small circles specify the log2FC values of each protein; the big circles are the average of the log2FC values, and columns indicate the sum of the log 2FC (**A**). Dynamic of identified differential enzymes in the palmitate biosynthetic (**B**) and saturated fatty acid elongation pathways (**C**). The pathway reconstruction was carried out by using the “cellular overview” tool in the Plant Metabolic Network (PMN) database platform.

## Data Availability

Not applicable.

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
