# Peer review of "Proteome Landscape during Ripening of Solid Endosperm from Two Different Coconut Cultivars Reveals Contrasting Carbohydrate and Fatty Acid Metabolic Pathway Modulation"

_ijms, 2023, doi:10.3390/ijms241310431_

Round 1

Reviewer 1 Report

The study was focused on proteome investigations during ripening of solid endosperm from two different coconut cultivars contrasting carbohydrate and fatty acid metabolic pathways modulation. The Authors revealed that proteomes related to glycolysis, photosynthesis, gluconeogenesis, and those associated with the biosynthesis and elongation of fatty acids, were up-accumulated in the solid endosperm of MPT, while in YGD they were down-accumulated. It supported that carbohydrate and fatty acid metabolisms differ among the developmental stages of the solid endosperm and between the dwarf and tall cultivars.

The paper is correctly prepared, the results obtained are quite interesting for a narrow circle of proteomic specialists. The results are mostly descriptive.

I suggest the following minor revisions:
- Introduction is overloaded in the content, it should be presented in a concise form,

- I suggest improving the resolution of the charts, as some of them are illegible to the reader (e.g. Figure 5, 7),

- Minor editing of English language required.

Minor editing of English language required.

Author Response

REVIEWER I:

I suggest the following minor revisions:

- Introduction is overloaded in the content, it should be presented in a concise form,

Author: We carefully revised the introduction and removed a couple of paragraphs whose lack do not harm the narrative of the manuscript. The deletion of these paragraphs is indicated by the Word corrector.

- I suggest improving the resolution of the charts, as some of them are illegible to the reader (e.g. Figure 5, 7),

Author: Figures 5 and 7 were downloaded from the program at higher resolution, and each was placed on a full horizontal page, improving readability.

- Minor editing of English language required.

Author: Misspelling words were corrected, and "s" was added to plural words as metabolisms and triacylglycerols (for example). Extra- spaces between words were removed, or introduced when two words were next to each other. I used  the Track Changes Tool in Microsoft Word to highlight the changes. Finally, the manuscript was revised and corrected by an English native Speaker.

Reviewer 2 Report

The goal of this manuscript is to present and discuss the significant role of chnges in the proteome during ripening of solid endosperm from two different Cocos nucifera L. cultivars.
All parts of the manuscript is interesting and clearly summarize new data valuable for the research community. The manuscript is well written and represents an important study that can offer insights for future research.

GENERAL COMMENTS:
TITLE
The paper title is well stated, it is informative and concise.

ABSTRACT, INTRODUCTION
Abstract is very well written with the key findings of the study. Introduction is concise, focused and informative.

MATERIAL AND METHODS
Material and research methods are presented appropriately and clearly. Experimental setup and the description in the methods section are well structured, and the statistical analysis is done alright.

RESULTS
The results obtained in this study are interesting. Results presented correctly.

DISCUSSION
In general, the discussion of results is correct and sufficient.

CONCLUSION
You have to re-write the conclusion because it is seemed as a repeat for the results without any key message. You have to wrap up your ideas and leave the reader with a strong final impression. I suggest to writing two other words on the aspect concerning the aspects where the future studies must be oriented.

LITERATURE
The items of literature included in the paper are rather sufficient and adequate to the subject of the paper.

I recommend revision of the English language by a native speaker or a commercial entity.

Author Response

REVIEWER 2:

CONCLUSION

You have to re-write the conclusion because it is seemed as a repeat for the results without any key message. You have to wrap up your ideas and leave the reader with a strong final impression. I suggest to writing two other words on the aspect concerning the aspects where the future studies must be oriented.

Author: The reviewer's comments were considered and the conclusion was modified to introduce a strong and conclusive message, as well as considerations about future studies that should be carried out.

I recommend revision of the English language by a native speaker or a commercial entity.

Author: Misspelling words were corrected, and "s" was added to plural words as metabolisms and triacylglycerols (for example). Extra- spaces between words were removed, or introduced when two words were next to each other. I used  the Track Changes Tool in Microsoft Word to highlight the changes. Finally, the manuscript was revised and corrected by an English native Speaker.